# ARAIM Stochastic Model Refinements for GNSS Positioning Applications in Support of Critical Vehicle Applications

**DOI:** 10.3390/s22249797

**Published:** 2022-12-13

**Authors:** Ling Yang, Nan Sun, Chris Rizos, Yiping Jiang

**Affiliations:** 1College of Surveying and Geo-Informatics, Tongji University, Shanghai 200092, China; 2School of Civil and Environmental Engineering, University of New South Wales, Sydney 2033, Australia; 3Department of Aeronautical and Aviation Engineering, The Hong Kong Polytechnic University, Hong Kong 999077, China

**Keywords:** global navigation satellite system (GNSS), integrity monitoring (IM), stochastic model, Gaussian overbounding, protection level (PL)

## Abstract

Integrity monitoring (IM) is essential if GNSS positioning technologies are to be fully trusted by future intelligent transport systems. A tighter and conservative stochastic model can shrink protection levels in the position domain and therefore enhance the user-level integrity. In this study, the stochastic models for vehicle-based GNSS positioning are refined in three respects: (1) Gaussian bounds of precise orbit and clock error products from the International GNSS Service are used; (2) a variable standard deviation to characterize the residual tropospheric delay after model correction is adopted; and (3) an elevation-dependent model describing the receiver-related errors is adaptively refined using least-squares variance component estimation. The refined stochastic models are used for positioning and IM under the Advanced Receiver Autonomous Integrity Monitoring (ARAIM) framework, which is considered the basis for multi-constellation GNSS navigation to support air navigation in the future. These refinements are assessed via global simulations and real data experiments. Different schemes are designed and tested to evaluate the corresponding enhancements on ARAIM availability for both aviation and ground vehicle-based positioning applications.

## 1. Introduction

One prospective application of ground-based global navigation satellite system (GNSS) positioning is autonomous driving within the context of intelligent transport systems (ITS). To be a key component in these systems, the accuracy, integrity, continuity and availability of GNSS positioning must be maintained at appropriate levels. Although high precision GNSS positioning technologies such as RTK (real time kinematic), PPP (precise point positioning), and PPP-RTK can achieve centimeter-level positioning accuracy, GNSS still faces serious challenges when applied to safety-of-life (SoL) applications [1].

Integrity monitoring (IM) should be conducted if GNSS positioning technologies are to be primarily used, and fully trusted, in ITS SoL applications. IM has two tasks [2]. Firstly, fault detection and exclusion (FDE) should be conducted to compensate for the positioning errors caused by significant system and observation faults [3]. Secondly, a warning should be given to the user within a preset time interval, known as the time-to-alert (TTA), if the true positioning errors (PEs) after FDE exceed a preset threshold called the alert limit (AL). Since the true PEs are not known in practice, the protection levels (PLs) are computed and compared with the AL instead. Theoretically, the PL statistically defines the largest PE that would happen without raising an alert, with a probability no higher than the permissible probability of hazardous misleading information (PHMI).

Originating from aviation applications, to satisfy the GNSS positioning accuracy, integrity, continuity and availability during different phases of flight, airborne-based augmentation systems (ABAS), satellite-based augmentation systems (SBAS) and ground-based augmentation systems (GBAS) have been developed since the mid-1990s [4]. Receiver autonomous integrity monitoring (RAIM) was initially proposed, as an ABAS technology solution, to support en route flight by only using GPS L1 pseudorange measurements [5,6]. To further support precise approach operations, SBAS and GBAS technologies have been developed and gradually standardized by the International Civil Aviation Organization (ICAO) [7]. These two types of augmentation technology rely on ground-based reference stations, firstly to monitor different types of threat that may result from large positioning errors, and secondly generate the integrity support message (ISM) for the airborne GNSS receivers [8,9,10]. With multiple next generation GNSSs and augmentation systems, the concept of advanced RAIM (ARAIM) is under study, where the burden of IM among space-based and ground-based augmentation systems and aircraft receivers is adjusted [4,11].

In general, IM for aviation is still based on the implementation of the standard GNSS single point positioning (SPP) mode, a snapshot least-squares (LS) procedure that uses pseudorange or carrier-smoothed code observations. In the ARAIM baseline algorithm, broadcast ephemeris, classical tropospheric and ionospheric delay correction/elimination models/methods are specified. As the operational environments are open sky, a mature and fixed stochastic model has been established to successfully compensate for the multipath and pseudorange noises for specific airborne GNSS receivers [4].

However, similar schemes that are used for ground vehicle-based positioning to maintain the required levels of accuracy, integrity, continuity and availability are lacking. Carrier phase-based RAIM (CRAIM) algorithms for RTK and PPP positioning using a Kalman filter have been studied for both float ambiguity or fixed ambiguity positioning modes [12,13,14]. El-Diasty [15] investigated the integrity of an autonomous real-time precise point positioning (PPP)-based GPS positioning system using the international GNSS service (IGS) real-time service (RTS) for maritime applications and applying the RAIM concept. More recently, there has been increasing interest in investigating RTK or PPP integrity on the basis of the ARAIM concept [16,17,18]. Most of the RTK and PPP integrity investigations have focused more on the ambiguity resolution process under nominal error conditions [19,20], and less on the integrity of the position solution under faulty conditions. In general, for vehicle-based GNSS positioning applications, IM comprehensively considering the following issues is still an open question:(1)Precise satellite orbit and clock error products have been widely used;(2)Advanced atmospheric delay correction models are available;(3)Receiver noises and environment-dominant multipath effects are quite diverse;(4)Multiple undetected pseudorange outliers, cycle slips, and uncalibrated system biases should be monitored in the fault modes;(5)FDE and PL calculation should be conducted during a Kalman filtering (KF) procedure instead of a snapshot LS procedure.

At the first stage, it is anticipated that the user-level integrity would be enhanced if a tighter and conservative stochastic model is applied. In this paper, the authors comprehensively refine the stochastic model and evaluate the potential enhancement of user-level integrity using the ARAIM concept, mainly for vehicle-based GNSS positioning applications. The rest of the paper is organized as follows. Firstly, an overview of the ARAIM concept as it was developed for aviation applications are given. Then, the prospective challenges of IM for vehicle-based GNSS positioning are discussed, with comparisons to aviation applications. Next, the three refinements of the ARAIM stochastic model for vehicle-based GNSS positioning are presented. Experiments and analyses to evaluate the ARAIM performance improvements follow. The concluding remarks of this study are presented last.

## 2. Overview of ARAIM Concept for Aviation

The ARAIM concept and its baseline algorithm are outlined in the GPS Evolutionary Architecture Study (GEAS) in the GEAS phase II report [21], which has been further developed within the working group C ARAIM technical subgroup (ARAIM TSG) [22]. Under this framework, a ground system is required to monitor the nominal performance and fault rates of the GNSS constellations in question. These integrity data are contained in the ISM that is generated on the ground and broadcast to the airborne users [22]. The target operational level for ARAIM is LPV-200, which is incompletely specified by ICAO standards additionally recommended practices (SARPs) [7]. ARAIM adds four main elements to the basic RAIM concept: multiple constellations, dual frequency, a deeper threat analysis and the possibility of updating the ISM gained by sufficient data and experience. For all of these reasons, ARAIM could significantly reduce a large portion of the SBAS operational budget, and is considered the primary IM method for future aviation [21]. It could also be a good basis and reference for investigation of IM for ground vehicle-based GNSS positioning applications.

### 2.1. ARAIM Algorithm Flowchart

A brief overview of the ARAIM baseline user algorithm proposed in the [22] is given in this subsection. The ARAIM algorithm flowchart is shown in Figure 1. Firstly, the nominal functional model and stochastic model are constructed based on the SPP mode. Then, all-in-view positions are solved using the nominal models, and a group of fault-tolerant positions is resolved in the monitored fault modes. A specific monitored fault mode assumes that one or more pseudoranges are faulty and should be excluded to obtain a fault-tolerant solution. Next, an FDE procedure, consisting of global hypothesis testing and multiple parallel local hypothesis testing, is used. The former is a Chi-Square test (CST) that depends on the observation residuals of the all-in-view solution, and the latter is a solution separation threshold test (SST) that depends on the differences between the all-in-view solution and a specific fault-tolerant solution. There are three criteria that follow the FDE procedure: (1) a specific fault mode is identified, and should be excluded from the all-in-view observation collection; (2) the actual failure is outside the collection of monitored fault modes; and (3) the current all-in-view observations are fault free. Only at the third decision point are the integrity-related parameters (PLs) and other accuracy and continuity-related parameters calculated and compared with the corresponding criteria defined by ICAO SARPs [7], upon which the availability of the ARAIM algorithm is determined.

According to Figure 1, the stochastic model that characterizes the nominal error in each observation would be an important factor influencing the performance of ARAIM algorithms. A general interpretation of the mathematical models used for positioning and IM under the ARAIM framework is given in next subsection, and followed by a more detailed discussion on the stochastic model.

### 2.2. Interpretation of the ARAIM Mathematical Model

In general, the linearized mathematical model for GNSS positioning contains a functional and a stochastic model
(1)$$y=Ax+e\;\mathrm{and}\;e\sim N\left(0,D_{ee}\right)$$
where ***y*** is the *m* × 1 vector of observed minus computed (OMC) values of pseudorange observations, ***x*** is the *t* × 1 vector of unknown parameters and ***A*** is the *m* × *t* design matrix determined by the satellite-to-receiver geometry. The error vector ***e*** contains all unmodeled errors and is modeled as zero-mean normal distributed random variables with a variance of $D_{ee}$.

It is noted that the covariance matrix $D_{ee}$ for IM must be conservative to ensure integrity. However, an overly conservative variance matrix would exaggerate PLs in the position domain and therefore reduce the system availability. Under the SPP mode, the random error vector e generally consists of three types of errors: the satellite-related errors, the residual atmospheric errors after correction using products or models and the receiver noises and multipath errors. For this reason, the covariance matrix, $D_{ee}$, is modeled as a sum of the three terms.

If dual-frequency observations are available, the ARAIM algorithm will use the ionosphere-free (IF) combination of pseudorange observations for positioning. For each pseudorange OMC value, $y_{i}$, the residual error is represented by a nominal Gaussian distributed random error, $e_{i}$, and a maximum bias, $b_{nom,i}$. The former is characterized by the stochastic models for positioning and IM and the latter is needed only for integrity, to compute a conservative and safe PL value, and not for positioning. For each pseudorange observation, the nominal error, $e_{i}$, is characterized by two variances: a conservative one used for integrity purposes, and a less conservative one used for accuracy and continuity purposes [4].

For advanced positioning modes such as RTK, PPP, or PPP-RTK, the tropospheric and/or ionospheric-related errors could be considered unknown parameters, and corresponding constraint equations are added to the set of observation equations [23,24]. In this case, apriority variances of the constraints would affect the performance of positioning and IM [25]. The stochastic models must adequately describe the uncertainties of the observations and constraints by being neither overly conservative nor optimistic.

### 2.3. ARAIM Stochastic Models to Characterize the Nominal Error of Pseudoranges

According to the ARAIM framework, the diagonal (co)variance matrices $D_{ee,int}$ (used for integrity) and $D_{ee,acc}$ (used for accuracy and continuity) for IF combination pseudorange observations are given by [4]
(2)$$\{\begin{array}{l} D_{ee,int}\left(i,i\right)=\sigma_{URA,i}^{2}+\sigma_{tropo,i}^{2}+\sigma_{user,i}^{2}\\ D_{ee,acc}\left(i,i\right)=\sigma_{URE,i}^{2}+\sigma_{tropo,i}^{2}+\sigma_{user,i}^{2} \end{array}$$
where ‘*i*’ is the satellite indicator; $\sigma_{URA,i}$ and $\sigma_{URE,i}$ are the satellite-related errors of satellite *i*, used for integrity as well as for accuracy and continuity, respectively; $\sigma_{trop,i}$ represents the residual tropospheric delay errors after model correction; and $\sigma_{user,i}$ represents the receiver noises and multipath errors.

The first terms in Equation (2) should characterize the signal in space (SIS) user range error (URE), which is dominated by satellite orbits and clock errors [26]. Traditionally, SIS URE is assumed to follow a zero-mean normal distribution with a standard deviation (STD) represented by the broadcast user range accuracy (URA), which is denoted by $\sigma_{URA,i}$ in Equation (2). The other one, $\sigma_{URE,i}$, used for accuracy and continuity purposes, is not safety critical; therefore, it will require a less conservative value and is usually set to be two thirds of the $\sigma_{URA,i}$ [27].

The second term, $\sigma_{trop,i}$, represents the residual tropospheric delay errors after model correction, and can be modeled as [4]
(3)$$\sigma_{trop,i}=\sigma_{ZPD}\times \frac{1.001}{\sqrt{0.002001+sin^{2}\left(E_{i}\right)}}$$
where $\sigma_{ZPD}$ is the residual Zenith Path Delay (ZPD) after model correction, and is set to be 0.12 [28]; and $E_{i}$ is the elevation angle of the satellite *i*, which projects the zenith $\sigma_{ZPD}$ to the line-of-sight direction.

The third term, $\sigma_{user,i}$, is the receiver-dependent error term. For GPS, it can be modeled as [4]
(4)$$\sigma_{user,i}^{2}=\frac{f_{L1}^{4}+f_{L5}^{4}}{f_{L1}^{2}-f_{L5}^{2}}\left(\sigma_{MP,i}^{2}+\sigma_{Noise,i}^{2}\right)$$
where $f_{L1}$ and $f_{L5}$ denote the frequencies of the GPS signals used to form the IF observation; and $\sigma_{MP,i}$ and $\sigma_{Noise,i}$ represent the multipath errors and receiver random noises, respectively. For airborne GPS receivers, these are modeled as
(5)$$\{\begin{array}{c} \sigma_{MP,i}=0.13+0.53\times e^{-\frac{E_{i}}{10}}\\ \sigma_{Noise,i}=0.15+0.43\times e^{-\frac{E_{i}}{6.9{}^{\circ}}} \end{array}$$

In Equations (1) and (2), the all-in-view positions and fault-tolerant positions are obtained via LS estimation, using which the PDE procedure and PL calculation are then performed.

## 3. Integrity Monitoring Challenges for Ground-Based GNSS Positioning

Although IM for aviation has been comprehensively investigated and three standard enhancement technologies have been announced by the ICAO, IM faces big challenges in ground-based GNSS positioning applications such as ITS. The different issues that IM should consider are summarized in Table 1. There are three issues to consider.

Firstly, different observations and positioning techniques are used. In aviation, the baseline IM algorithms are defined based on the assumption that a snapshot weighted LS position algorithm is implemented. Usually, differential or corrected pseudorange observations from GPS or GPS/Galileo dual systems are used. On the other hand, for ground-based positioning applications, multi-frequency and multi-constellation carrier phase observations are increasingly implemented using the RTK, PPP or PPP-RTK modes, seeking higher accuracy, integrity, continuity and availability. In such circumstances, positioning and IM should be based on a KF algorithm, which makes use of observations from all previous epochs. In theory, rigorous IM based on a KF algorithm, including the FDE procedure and PL calculation, should consider potential failure threats to both pseudorange and carrier phase observations.

Secondly, stochastic models for positioning and IM for ground-based positioning need to be redefined. In aviation, stochastic models only need to characterize the errors of pseudorange observations. As mentioned above, the total error in a raw or corrected pseudorange observation is the sum of satellite-related, atmospheric-related and receiver-related errors. The variances of these error terms have been explicitly defined by specifications under GBAS, SBAS or RAIM/ARAIM frameworks, where broadcast ephemerides, as well as the specific way to correct tropospheric and ionospheric delays, should be implemented. The observations are generated by an airborne receiver that satisfies the requirements of specified accuracy conditions, and should be operated under the assumed interference environment. Therefore, the receiver-related error term is characterized by a fixed and well-defined model. In contrast, for ground-based positioning, a variety of augmentation approaches can be implemented, i.e., using IGS precise satellite orbit and clock error products, eliminating or correcting the tropospheric and ionospheric delays by various advanced models and estimating those residual delays together with the navigation parameters. In addition, both the receiver manufacturers and the operational environments are diverse, so a single fixed model cannot characterize the receiver-related error term.

Thirdly, fault events and modes that need to be monitored are more complicated for ground-based positioning. Within the RAIM/ARAIM framework, the error in a pseudorange OMC value is divided into two parts: the unmonitored maximum nominal bias in the null hypothesis model for the fault-free mode, and the monitored unknown large bias in each alternative hypothesis model for the fault mode. Only satellite and constellation failures that cause the unknown large bias in position solutions are monitored with an assumed small occurrence rate (i.e., <10^−5^ for satellite, and <10^−4^ for constellation). For ground-based positioning, since both pseudorange and carrier phase observations would be used by various advanced positioning algorithms, modeling and monitoring the satellite-related and atmosphere-related errors becomes increasingly complicated. In addition, the effects caused by different types of receivers and the diversity of near-ground operational environments complicate matters further [29]. In terms of effects on observations, these uncertainties could result in data interruption, or larger pseudorange outliers and carrier phase cycle slips with higher occurrence rates, which in turn reduce the system availability.

## 4. Three Refinements for ARAIM Stochastic Models

Current stochastic models used in ARAIM baseline algorithms are only specified for aviation applications and are overly conservative in order to guarantee security. In this section, the stochastic models for ground vehicle-based GNSS positioning are refined thusly:(1)Gaussian bounds of IGS RTS products are used to consider the satellite effects delicately;(2)A variable standard deviation for characterizing the residual tropospheric delay is adopted to consider the atmosphere effects delicately;(3)An elevation-dependent model describing the receiver-related errors is adaptively refined to consider the receiver effects delicately.

In this study, the authors discuss these issues for the SPP mode. However, these investigations can also contribute to advanced-mode GNSS positioning, which will be undertaken in future work.

### 4.1. Gaussian Bounds to Characterize the Satellited-Related Error Term

For real-time precise positioning and navigation, the satellite precise orbit and clock error products provided by the IGS RTS can be applied. In this subsection, the variance of the satellite-related error term, denoted by $\sigma_{URA,i}$ in Equation (2), is estimated for this type of product. The baseline ARAIM algorithm assumes that $\sigma_{URE,i}=\frac{2}{3}\sigma_{URA,i}$, therefore only the value of $\sigma_{URA,i}$ that characterizes the SIS UREs of the IGS RTS products is investigated.

#### 4.1.1. Methodology

It would be inappropriate that $\sigma_{URA,i}$ be simply set to be the STD of the error samples, as the true distribution of the errors could be arbitrary and is usually non-Gaussian. For IM, the true distribution is usually replaced by a simpler overbounding distribution; $\sigma_{URA,i}$ is set to be the STD of this overbounding distribution.

By subtracting the RTS orbit and clocks from their post-processing counterparts, error samples for orbit and clock are obtained; these are denoted as $e_{\mathrm{orb},\;\mathrm{RTS}}$ and $e_{\mathrm{clk},\mathrm{RTS}}$. Subsequently, a much more conservative estimation of $\sigma_{URA,i}$ can be simply determined by
(6)$$\{\begin{array}{l} \sigma_{URA,i}^{2}={\left(\sigma_{\mathrm{orb},\mathrm{RTS}}^{i}\right)}^{2}+{\left(\sigma_{\mathrm{clk},\mathrm{RTS}}^{i}\right)}^{2}\\ m_{URE,i}=\left|m_{\mathrm{orb},\mathrm{RTS}}^{i}\right|+\left|m_{\mathrm{clk},\mathrm{RTS}}^{i}\right| \end{array}$$
where $m_{\mathrm{orb},\mathrm{RTS}}^{i}$ and $\sigma_{\mathrm{orb},\mathrm{RTS}}^{i}$ denote, respectively, the mean and STD values of the Gaussian overbounding distribution for $e_{\mathrm{orb},\mathrm{RTS}}^{i}$; and $m_{\mathrm{clk},\mathrm{RTS}}^{i}$ and $\sigma_{\mathrm{clk},\mathrm{RTS}}^{i}$ denote, respectively, the mean and STD values of the Gaussian overbounding distribution for ${\tilde{e}}_{\mathrm{clk},\mathrm{RTS}}^{i}$. The mean value, $m_{URE,i}$, can be attributed to the bias term, $b_{nom,i}$, as discussed above.

It is noted that Equation (6) does not follow the stringent definition of URA and URE in order to ensure the estimation is more conservative. It has been verified that the integrity risk computed using the convolution of the overbounding distributions described above is an upper bound of the integrity risk computed using the convolution of the true distributions. This means the PLs computed by using the overbounding mean and STD values are still conservative and can satisfy the integrity requirement [30].

#### 4.1.2. Dataset and Results

There are several IGS RTS products that can be received in real time. In this study, the IGC01 products are used. The IGC01 products are firstly generated by individual real-time analysis centers (ACs) and combined by the European Space Agency’s Space Operations Center (ESA/ESOC). In particular, IGC01 is a single-epoch combined product with an update rate of 5 s for both orbit and clock [31]. The IGS final products, with update rates of 15 min/30 s for orbits/clocks, are used as the reference for the computation of the nominal accuracy. To avoid the effect of interpolation, a comparison interval of 15 min is used. The dataset used for evaluating $\sigma_{URA,i}$ corresponds to the period 1 September 2019 to 20 September 2020. Only GPS satellites were considered.

As an example, the $e_{\mathrm{orb},\;\mathrm{RTS}}$ and ${\tilde{e}}_{\mathrm{clk},\mathrm{RTS}}^{i}$ values for satellite G02, obtained by the preprocessing procedure described above, are shown in Figure 2. These residuals are then used as inputs for the aforementioned two-step overbounding estimation technique. Figure 3 shows the corresponding probability distribution function (PDF) and CDF of the $e_{\mathrm{orb},\;\mathrm{RTS}}$ samples (blue), the symmetric unimodal bounding distribution after the first step (red), and the final Gaussian overbounding distribution after the second step (green).

The mean and STD values of $e_{\mathrm{orb},\;\mathrm{RTS}}$ and ${\tilde{e}}_{\mathrm{clk},\mathrm{RTS}}^{i}$ for each GPS satellite, directly obtained from the green lines in Figure 3, are again plotted in Figure 4, with blue bars and red line segments indicating the mean and STD values, respectively. Combining the effects of orbits and clocks using Equation (6), the mean and STD values for SIS URE of each GPS satellite further contribute to the integrity parameters of $b_{nom,i}$ and $D_{ee}\left(i,i\right)$, and are then used to compute the conservative and safe PLs in the position domain.

From Figure 4, one can see that the mean and STD values for orbits and clock errors of different satellites can vary significantly. If a common value for all satellites is to be adopted, the most conservative one should be chosen. In this case, the computed PLs in the position domain will also be overly conservative and, hence, reduce the system availability.

### 4.2. A Variable Standard Deviation to Characterize the Residual Tropospheric Delay

The current tropospheric delay correction model used in ARAIM is the one recommended by the Radio Technical Committee for Aeronautics (RTCA) and is described in RTCA’s Minimum Operational Performance Standards (RTCA MOPS). It states that, in terms of STD, the maximum ZPD residuals after model correction is 0.12 m [28]. However, this constant value does not consider the accuracy variations of different seasons and user latitudes. When applying another tropospheric correction model, such as the GPT2/GPT2w [32], a much smaller STD would be expected. In this subsection, the general extreme value (GEV) analysis method is used to evaluate the variable $\sigma_{ZPD}$, considering the geographical and seasonal variations of the ZPD residuals after model correction [33].

#### 4.2.1. Methodology

For integrity, $\sigma_{ZPD}$ should characterize the maximum ZPD residuals after model correction. Variable $\sigma_{ZPD}$ for a tropospheric delay correction model can be established via the following six steps [33].

*Step 1:* The residual series δ is obtained by subtracting the model estimated ZPDs from the reference values. Since the hydrostatic and wet components of ZPD usually have different characteristics, the residual series can be divided into these two components, denoted by $\delta_{h}$ and $\delta_{w}$.

*Step 2*: The seasonal variations of δ are extracted by calculating the daily STDs of the residuals at all grid points for each latitude band of 10°. Then, a periodic function is used to fit the daily STD series, σ(DOY), so as to represent this seasonal variation.
(7)$$\sigma \left(DOY\right)=A_{0}+A_{1}\cos\left(\frac{DOY-DOY_{0}}{365.25}2\pi \right)+A_{2}\sin\left(\frac{DOY-DOY_{0}}{365.25}2\pi \right)$$

*Step 3:* The annual maximum and minimum of the normalized residual series $\tilde{\delta }=\delta /\sigma \left(DOY\right)$ are used to fit the GEV distribution, respectively, and the obtained fitted parameters are used to calculate the extreme quantiles. The largest absolute value of the two quantiles is chosen as the extreme maximum value ${∆}_{n,max}$, with *n* being the band indicator.

*Step 4:* To completely overbound the original residuals δ, an offset parameter ${∆}_{0}$ is calculated using another GEV distribution which fits the maximum and minimum of the daily mean value of the residual series δ. Then, the upper bound of the absolute tropospheric delay residuals is given by
(8)$${∆}_{max}\left(DOY,band\right)={∆}_{0}+\sigma \left(DOY\right)\times {∆}_{n,max}$$
which is a periodic function with parameters of ${∆}_{0}$; ${∆}_{n,max}$ from a GEV distribution; and $A_{0}$, $A_{1}$, $A_{2}$, $A_{3}$ from a periodic fitting function for σ(DOY).

*Step* 5: A variable standard deviation, $\sigma_{max}\left(DOY,band\right)$ of the original residuals δ is obtained by converting the value of ${∆}_{max}\left(DOY,band\right)$ to its STD value
(9)$$\sigma_{max}\left(DOY,band\right)=\frac{\left|{∆}_{max}\left(DOY,band\right)\right|}{K}$$
where *K* is the right tail quantile of the PDF of a standard normal distribution at the extreme probability level, corresponding to the preset PHMI.

*Step 6:* The total $\sigma_{ZPD}$ used for positioning and IM combines the hydrostatic and wet components
(10)$$\sigma_{max,ZPD}^{2}\left(DOY,band\right)=\sigma_{max,h}^{2}\left(DOY,band\right)+\sigma_{max,w}^{2}\left(DOY,band\right)$$
where $\sigma_{max,h}\left(DOY,band\right)$ and $\sigma_{max,w}\left(DOY,band\right)$ are the STDs calculated by Equation (9), which overbound the hydrostatic ($\delta_{h}$) and wet ($\delta_{w}$) residuals in each latitude band, respectively. For IM, the variable value of $\sigma_{max,ZPD}\left(DOY,band\right)$ will replace the constant value of $\sigma_{ZPD}=0.12\;m$ in Equation (9).

#### 4.2.2. Dataset and Results

Two commonly used tropospheric delay correction models, UNB3 and GPT2w, were evaluated. For a worldwide integrity analysis, the grid-wise VMF3 products with a resolution of 1 × 1 degree between 2000 to 2016 were selected as the reference.

As an example, considering the zenith hydrostatic delays (ZHD) residuals of GPT2w in the latitude band 41° N–50° N, the original residual series (blue dots) and its upper bounds (red periodic curves) are shown in Figure 5. It can be seen that the ZHD residual series has an obvious seasonal variation. This seasonal variation is preserved by fitting the daily STD series of the residuals with a periodic function. Consequently, the upper bounds can successfully overbound the residuals.

To further investigate the effectiveness of the upper bounds, the Stanford diagrams of the two models are shown in Figure 6. It can be seen that all points fall above the corresponding diagonal. By taking the absolute residuals as the horizontal axis and the corresponding upper bound values as the vertical axis, these results indicate that the residuals can always be bounded, which verifies the effectiveness of the upper bound values. Furthermore, one can see that the upper bounds of GPT2w are significantly smaller than those of the UNB3 model. This is consistent with our knowledge of the accuracies of the two models.

The variable STDs for the two models, in each latitude band, are plotted in Figure 7. This confirms the fact that a constant value of 0.12 m is conservative enough for the UNB3 model. When GPT2w is used, the constant value can be further reduced to 0.08 m. In general, the values show significant discrepancies at different latitude bands and different seasons. The STDs vary between 0.05 m and 0.12 m for the UNB3 model, and between 0.03 m and 0.08 m for the GPT2w model. Therefore, a constant value would lead to an overly conservative estimation of the PLs in the position domain. The UNB3 model performs much better in the northern hemisphere, where the STDs are typically smaller than 0.1 m, and usually increase during summer. In low latitude regions of the southern hemisphere, the STDs would reduce to 0.06 m during summer. In contrast, the GPT2w model exhibits better performance in the southern hemisphere, and opposite seasonal variations are observed in high and low latitude regions. For the GPT2w, the residual STDs would reach a minimum during winter in high latitude regions, while in equatorial regions, the STDs during winter are usually much larger than those during other seasons.

### 4.3. An Elevation-Dependent Model with Adaptive Coefficients to Characterize the Receiver-Related Errors

According to the ARAIM framework, the receiver-related error term is characterized by an elevation-dependent model with constant coefficients, as given in Equation (5), which can only represent the errors of an airborne GPS receiver when it operates under a specific interference environment. For ground-based positioning, such a fixed model would be inadequate, if different types of receivers are used under diverse operational environments. In this subsection, the elevation-dependent model to describe the receiver-related errors is adaptively refined using least-squares variance component estimation (LSVCE), with the variances of satellite-related and atmospheric-related errors being considered as known parts.

#### 4.3.1. Methodology

There are many elevation-dependent models that can be used to characterize receiver-related errors with adaptive coefficients. The authors adopt the conventional one [34].
(11)$$\sigma_{user,i}=\frac{a}{b+sin\left(E_{i}\right)}$$
where *a* and *b* are the model coefficients that need to be estimated. Other variables have been defined in previous section.

LSVCE is conventionally used to construct a more accurate and adaptive (co)variance matrix based on the LS criterion. According to LSVCE, the functional and stochastic models are processed in a unified manner, and the variance matrix can be customized by users to adapt to different applications.

The (co)variance matrix of the linear Gauss–Markov model (Equation (1)) can be represented as [35]
(12)$$D_{ee}=D_{0}+\overset{p}{\underset{k=1}{\sum }}\sigma_{k}D_{k}$$
where $D_{0}$ is the known part of the (co)variance matrix, which combines the satellite-related and atmospheric-related errors as given above; $\sigma_{k}$ is the (co)variance component; and $D_{k}$**,**
k=1,⋯,p, is the cofactor matrix that determines the position of the (co)variance $\sigma_{k}$ in the matrix.

Combining Equations (1) and (12), the mathematical model of LSVCE contains two groups of unknowns: the original parameter vector x and the (co)variance component $\sigma_{k}$, k=1,⋯,p. The LS estimate of x and e is
(13)$$\left[\begin{array}{c} \hat{x}\\ \hat{e} \end{array}\right]=\left[\begin{array}{c} D_{\hat{x}\hat{x}}A^{T}D_{ee}^{-1}\\ R^{T} \end{array}\right]y$$

Using the law of error propagation, the expectation and (co)variance of the estimate are
(14)$$E\left\{\left[\begin{array}{c} \hat{x}\\ \hat{e} \end{array}\right]\right\}=\left[\begin{array}{c} x\\ 0 \end{array}\right],\;D\left\{\left[\begin{array}{c} \hat{x}\\ \hat{e} \end{array}\right]\right\}=\left[\begin{array}{cc} D_{\hat{x}\hat{x}} & 0\\ 0 & R^{T}D_{ee}R \end{array}\right]$$
where E{·} and D{·} are the mathematical expectation operator and dispersion operator, respectively; $D_{\hat{x}\hat{x}}={\left(A^{T}D_{ee}^{-1}A\right)}^{-1}$ is the estimated (co)variance matrix of $\hat{x}$; and $R^{T}=I-AD_{\hat{x}\hat{x}}A^{T}D_{ee}^{-1}$ is the idempotent matrix that projects the observation error e onto its estimate $\hat{e}$.

Substituting Equation (12) into Equation (14)
(15)$$E\left\{\hat{e}{\hat{e}}^{T}-R^{T}D_{0}R\right\}=\sum_{k=1}^{p}\sigma_{k}R^{T}D_{k}R$$

By using the *vh*-operator, the matrix observation equation Equation (15) is transformed to the familiar vector–matrix form of the linear observation equation
(16)$$E\left(y_{\mathrm{vh}}\right)=A_{\mathrm{vh}}\sigma$$
where $y_{\mathrm{vh}}=\mathrm{vh}\left(\hat{e}{\hat{e}}^{T}-R^{T}D_{0}R\right)$; $A_{\mathrm{vh}}=\left[\mathrm{vh}\left(R^{T}D_{1}R\right),\ldots ,\mathrm{vh}\left(R^{T}D_{p}R\right)\right]$; and $\sigma ={\left[\sigma_{1},\ldots ,\sigma_{p}\right]}^{T}$. The weighted LSVCE estimate of σ is
(17)$$\hat{\sigma }={\left(A_{\mathrm{vh}}^{T}W_{\mathrm{vh}}A_{\mathrm{vh}}\right)}^{-1}A_{\mathrm{vh}}^{T}W_{\mathrm{vh}}y_{\mathrm{vh}}$$
with the weight matrix $W_{\mathrm{vh}}$ being defined as
(18)$$W_{\mathrm{vh}}=D^{T}D_{ee}^{-1}⊗D_{ee}^{-1}D$$
where ⊗ denotes the *Kronecker product*; and D is the *duplication matrix* that satisfies the condition Dvh(·)=vec(·). For a definition and description of the properties of the *vh*-operator, *vec*-operator and *duplication matrix* D, the reader can refer to Appendix A [35].

Assuming n observations, their corresponding (co)variance components $\sigma_{k}$ and k=1,⋯,n are estimated via Equations (15)–(18). These estimated (co)variances are then substituted into Equation (11) to fit the model coefficients. To ensure the redundancy of LSVCE, the true coordinates from the SINEX file provided by IGS were used, and only the receiver clock error was estimated as the unknown parameter.

Finally, by checking whether a set of samples matches the assumed PDF, the reliability of the stochastic model can be evaluated [36]. The test statistic *T* constructed by the estimated unknown parameter $\hat{x}$ and its (co)variance matrix $D_{\hat{x}\hat{x}}$ is
(19)$$T={\left(\hat{x}-\tilde{x}\right)}^{T}D_{\hat{x}\hat{x}}^{-1}\left(\hat{x}-\tilde{x}\right)\sim \chi^{2}\left(t\right)$$
where $\tilde{x}$ is the true value of the parameter vector. If the stochastic model is accurate, the test statistic *T* should follow a centralized $\chi^{2}$ distribution with a freedom of *t*. For GNSS positioning applications, the authors have only used the 3×1 positioning parameter vector for testing.

#### 4.3.2. Dataset and Results

The GPS dual-frequency IF combination pseudorange observations, from four IGS GNSS continuously operating receiver stations at different latitudes, were used to fit the coefficients and to evaluate the reliability of the stochastic model. The time period of the data was between 5th September 2019 and 4th October 2019.

For each pseudorange observation, the variances of the first two terms at the right side of Equation (2) are considered the known part ***D***_0_, using IGC01 products and GPT2w tropospheric delay corrections. The variances necessary to characterize the receiver-related error term are estimated with LSVCE and are used to fit the model coefficients of Equation (14). The fitting results are listed in Table 2, against by the IGS station’s name and latitude. The variation in values of the two model coefficients for different stations can be seen. The value of coefficient *a* varies between 0.4 and 0.8, and the value of coefficient *b* varies between 0.02 and 0.15. It should be noted that the two coefficients have the smallest values at the station NYA2, which is located in the Arctic region.

To evaluate the reliability of the adaptive elevation-dependent model, the test statistic *T*, as given in Equation (19), was calculated. The elevation-dependent model with constant coefficients applied in GAMP [37] was used for a comparison
(20)$$\sigma_{user,i}=3\sigma_{0}\sqrt{1+\frac{1}{sin^{2}\left(E_{i}\right)}}$$
where $\sigma_{0}$ is adopted as 0.3 m for pseudorange observations, and the multiplication factor 3 is consistent with the IF combination.

Figure 8 shows the probability histograms for the statistic *T* (blue bars), together with the PDF of the $\chi^{2}\left(3\right)$ distribution (red curves) before and after the stochastic model refinement for the receiver-related error term. Before refinement, Equation (20) with constant coefficients is used to characterize the third term on the right side of Equation (2). After refinement, Equation (11), with adaptive coefficients listed in Table 2, is used. As the common parts, the satellite-related and atmospheric-related terms are defined by Equation (7) and Equation (10) using the IGC01 products and GPT2w model, respectively.

The upper panels are plots of the results for the four IGS stations before refinement, and the lower panels are plots of the corresponding values after refinement. The upper panels show that the statistics are mainly distributed on the left side of the PDF of $\chi^{2}\left(3\right)$ distribution, which indicates that the (co)variance matrix is overly conservative before refinement. The lower panels show that the distributions of the statistic *T* in general have better consistency with the centralized $\chi^{2}\left(3\right)$ distribution. This confirms that the stochastic models at these four IGS stations have become more reasonable after refinement.

## 5. Experiments and Discussion

In this section, the performance enhancements for aviation and ground-based positioning applications are further investigated. The open-source Matlab Availability Analysis Simulation Tools (MAAST) released by the GPS Lab, Stanford University, are used for the ARAIM evaluation. Only the GPS constellation is considered. To ensure consistency with the values estimated in last section, the broadcast ephemeris and the IGC01 RTS products, over the same time span between 1 September 2019 and 20 September 2020, were used for the integrity analyses. The required integrity is 10^−7^, and prior probabilities of a satellite or constellation fault were set to be 10^−5^ and 10^−4^, respectively. In both applications, improvements on ARAIM availability and PLs were analyzed using different options for stochastic model refinements. As the refinement for the receiver-related error term should rely on real data, the corresponding performance improvements are evaluated using the actual observations collected by the aforementioned IGS stations.

### 5.1. Worldwide ARAIM Performance Prediction

Seven schemes are tested to demonstrate the performance improvements brought about by refining the stochastic model of the three error terms for the aviation and ground-based positioning applications.

As the fixed model given by Equations (4) and (5) is only specified for avionics receivers, the receiver-related error term should be redefined. Detailed designs for parameters for the seven schemes are given in Table 3, where Schemes A–C are designed for aviation and Schemes D–G are designed for ground-based positioning. The cut-off elevation angle is set to be 5 degrees, and other constant parameters required for the ARAIM algorithm are the same as those given in [4].

For aviation, the differences among Schemes A–C are:

Scheme A—Original design of the baseline algorithm: $\sigma_{URA,i}$ for all satellites is set to be 1 m, $\sigma_{trop,i}$ is defined by Equation (3), with $\sigma_{ZPD}$ being a constant value of 0.12 m, and $\sigma_{user,i}$ is specified by Equations (4) and (5).

Scheme B—Refinement using IGS RTS products (IGC01): $\sigma_{URA,i}$ for each satellite is defined by Equation (6), and the other error terms are the same as those in Scheme A.

Scheme C—Refinement using IGS RTS products (IGC01) and spatiotemporal-varying STDs for residual tropospheric delays: $\sigma_{URA,i}$ for each satellite is determined by Equation (6), $\sigma_{ZPD}$ is specified by Equation (10) and $\sigma_{user,i}$ is the same as for Scheme A and B.

Schemes D–F are the counterparts of A–C for the ground-based positioning applications, where the stochastic model of the receiver-related error term is now defined by Equation (20). Scheme G is intended for the evaluation of the ARAIM performance improvement brought about by using a more accurate tropospheric correction model, via a comparison with the results of Scheme F.

#### 5.1.1. Results and Discussion for Aviation Application

The global coverage of ARAIM availability satisfying the LPV-200 requirements for aviation under Schemes A, B and C are shown in Figure 9, with each point representing a 1 × 1-degree grid value at 50 m altitudes. Each point is calculated every 10 min to obtain the availability over the entire time span of 386 days. It shows that under the original design of the baseline algorithm, 99.5% availability (defined as the percentage of time that the availability criteria are met at the user grid point), coverage is just 22.4%, while the corresponding percentages can reach 91.05% and 94.62% availability under Scheme B and C, respectively. This indicates that a huge improvement of ARAIM availability can be achieved when the IGS precise orbit and clock products are used, and a further 3.5% improvement is possible by using a time-varying STD value for the atmospheric-related error term. Under Scheme A, ARAIM availabilities higher than 99.5% are mainly possible in the equatorial and mid-latitude regions, and the percentages reduce to 95% in the regions at 20° N–40° N and 20° S–40° S. The worst performance is observed in the polar regions, where the availability percentages reduce to 50%. Under Scheme B and C, 99.5% availability is possible in most regions. Even in the polar regions, the availability percentages are significantly improved and can be higher than 95%.

The 99.5% VPL maps under the three schemes are further plotted in Figure 10. These maps show a significant improvement from Scheme A to B, when IGS RTS products are used instead of the broadcast ephemeris. Under Scheme A, 99.5% of VPLs in low and mid-latitude regions are mainly between 20 m and 35 m; those values in the polar regions are much larger, being between 40 m and 50 m. Maximum values, larger than 50 m, although rare, are observed in Antarctica. Under Scheme B, more than 50% of these values in low and mid-latitude regions are reduced to less than 20 m, and the majority of those values in the polar regions are between 20 m and 35 m. By further applying spatiotemporal-varying STDs for residual tropospheric delays under Scheme C, VPLs are generally reduced across the globe. The coverage where 99.5% VPLs are lower than 15 m is expanded in mid-latitude regions. Improvements are also observed in the equatorial and Arctic regions, where the coverage of 99.5% VPLs less than 20 m is significantly increased. On the other hand, there are fewer improvements in the Antarctic, which is coincident with those values in Figure 7, as the spatiotemporal-varying value of *σ*_ZPD_ in this region has less reduction compared with the constant value of 0.12 m, which is used under Scheme B.

Similarly, the 99.5% HPL maps under the three schemes are shown in Figure 11. One can see that 99.5% HPLs are almost less than 40 m across the globe under Scheme A, and smaller values, less than 20 m, are observed in equatorial and polar regions. Under Scheme B, those values are almost less than 20 m, which indicates a significant improvement on HPLs in mid-latitude regions. In general, little difference is observed between Scheme B and C in the current color bar legend. It is anticipated that more detailed differences would appear if the color bar interval was narrower. However, this uniform color bar setting is coincident with the horizontal alert limit (HAL) criterion from the ICAO SARPs.

In general, the results and analyses in this section show that worldwide ARAIM availability for aviation can be significantly improved by using the IGS RTS products and a tighter and variable value for *σ*_ZPD_. Compared with HPLs, the improvement to VPLs is more significant, as the AL for VPL is generally more stringent for aviation.

#### 5.1.2. Results and Discussion for Ground-Based Positioning Application

One expects similar improvements to be observed for ground-based positioning. Compared with vertical positioning (for aviation applications), horizontal positioning is more crucial for ground-based positioning users. The 99.5% HPL maps for Schemes D–G are given in Figure 12, where a smaller color bar interval and range are chosen to show more details. Among these four schemes, smaller values are always observed in the equatorial and polar regions, and the largest values always appear in the regions around 60° N and 60° S. This indicates that horizontal ARAIM availability in the equatorial and polar regions is higher than in mid-latitude regions. Under Scheme D, those values are generally between 15 m and 30 m in the equatorial and polar regions, and between 30 m and 40 m in mid-latitude regions. Under Scheme E, those values are generally reduced by 5 m to 10 m globally. This improvement is attributed to using IGS RTS products instead of the broadcast ephemeris. Discrepancies among Schemes E–G are marginal, which indicates that a less conservative $\sigma_{ZPD}$ to characterize the residual tropospheric delay has an insignificant role in enhancing horizontal ARAIM availability.

To further highlight the discrepancies among Scheme E, F and G, the HPL mean values at each user grid point are plotted along with the point number in Figure 13, where the user grid points are numbered from [90° S, 180° W] to [90° N, 180° E], sorted by longitude at each latitude. Figure 13 shows that the discrepancies among the three schemes are mostly at the centimeter level, and the mean HPLs under Scheme G are always the smallest at each user grid point. This is coincident with those values of $\sigma_{ZPD}$ shown in Figure 7, where $\sigma_{ZPD}$ differences between UNB3 and GPT2w are at the centimeter level. In general, the receiver-related error is the dominant term of the stochastic model for ground-based positioning; therefore, centimeter differences in $\sigma_{ZPD}$ under Scheme E, F and G only lead to small variations in the HPLs. Discrepancies among the three schemes become more obvious in mid-latitude regions, where the HPLs are also larger.

The above analyses assume that the user is in an open-sky environment, which is a condition that usually cannot be satisfied in ground vehicle-based positioning applications. To demonstrate the performance degradation caused by receiving ambient occlusion, Schemes H, I and J are designed with the cut-off elevation angle being 10, 15 and 20 degrees, while the other parameters are kept the same as those of Scheme G. The 99.5% HPL maps of the three schemes are shown in Figure 14, for comparison with the results of Scheme G in Figure 13. One can see that HPLs are globally larger than 120 m when the cut-off elevation angle is increased to 20°. Even for a cut-off elevation angle of 15°, HPLs are still larger than 120 m across nearly two-thirds of the globe, with smaller values mainly located in the equatorial regions. A cut-off elevation angle of 10° can ensure a value smaller than 50 m in more than 90% of the world. The results in Figure 14 indicate that satellite visibility is a crucial factor in ARAIM availability. For ground-based positioning, especially in city canyon areas, accurate prediction of ARAIM performance should rely on the real data or use an accurate 3D city model for the prediction of signal occultations.

### 5.2. Local Site-Wise Assessment Based on Real Data

The receiver-related error term of the stochastic model should characterize the receiver noises and multipath errors, and therefore the model should be able to accommodate different receivers and operational environments. Via LSVCE, receiver-related errors can be optimally modeled by an elevation-dependent function with adaptive coefficients. To evaluate the ARAIM performance improvements via this refinement, the same dataset as that in last section is used. To analyze the impacts of fault detection and exclusion, we simulated 20 m gross errors on two pseudoranges at each station during the 1000th and 2000th epoch. The Stanford diagrams of horizontal positioning errors (HPEs) and HPLs before and after the refinement at the four IGS stations are shown in Figure 15, with HAL being 40 m. The HPEs are obtained by subtracting the estimated horizontal coordinates from the truth values, which are given by the SINEX file provided by the IGS. The upper panels are the results at the four IGS stations before refinement, under Scheme G, where Equation (20) with fixed coefficients is used to characterize the receiver-related errors. The middle and lower panels are corresponding results after refinement, without and with the FDE, where Equation (11) with adaptive coefficients, as listed in Table 2, is used to characterize the receiver-related errors, and the other parameters are the same as those of Scheme G.

In Figure 15, the top and middle panels generally show that HPLs at the IGS stations are significantly reduced and more concentrated after refinement, while the HPEs show less improvement. This indicates that stochastic model refinement of the receiver-related error term mainly contributes to improving the ARAIM availability rather than the positioning accuracy. Before refinement, the HPLs at the IGS stations are mainly larger than 20 m, and there are significant instances where the HPLs exceed the HAL, which leads to system unavailability. After refinement, HPLs at the IGS statins mainly are in the range of 10 m to 30 m. From another aspect, when PLs reduce, points in these Stanford diagrams generally move down, and the number of MI and HMI epochs increases accordingly. In this case, misleading or dangerous misleading may occur with higher probabilities, so FDE is necessary. Results in the middle and bottom panels further show the performances of the MHSS test at the four stations after stochastic model refinement. It shows that after refining the stochastic model, MHSS tests can successfully detect and exclude the faulty pseudoranges and avoid misleading.

There are also significant differences in performance at different stations. At the polar station NYA2, HPEs are obviously smaller than HPEs at the other three stations, and the HPLs reduce to a much lower range of 10 m to 25 m after refinement. The worst performance is at the mid-latitude station DAEJ, where the HPLs before and after refinement are mainly in the range 25 m to 50 m, and 10 m to 40 m, respectively. Slightly better performance is observed at the station REUN, where the HPLs after refinement are reduced to a range of 10 m to 30 m, while the values before refinement are mainly between 20 m and 50 m.

In general, the results evaluated via real data shown in Figure 15 are coincident with those prediction results in Figure 12. All of these results indicate that ARAIM availabilities of the GPS constellation are typically higher in the equatorial and polar regions, and significantly lower in mid-latitude regions (around 40 to 60 degrees’ latitude).

Table 4 lists the number of total observed epochs, the number (and percentages) of ARAIM availability epochs before and after the stochastic model refinement, as well as the number of MI and HMI epochs without and with FDE after stochastic model refinement, at the four IGS stations. A total of 26,000 epochs are tested at each station. Before refinement, there are significant latitudinal discrepancies, and higher ARAIM availabilities appear at stations in the equatorial and polar regions. The most unavailable epochs are at the mid-latitude station DAEJ, where the percentage of unavailable epochs is 14.715%. After the receiver-related error term of the stochastic model is refined, the unavailable percentages are all significantly reduced to 0.0%. This confirms that the receiver-related error is one of the dominant factors that influence system availability. Furthermore, the number of MI and HMI epochs also reduce to zero when FDE is adopted after stochastic model refinement. This further proves that stochastic model refinement also enhances the FDE capability.

According to the ARAIM baseline algorithm, the first order ionospheric delays can be corrected using the dual-frequency IF combination, and the residual tropospheric delays after model correction are usually at the centimeter level. For aviation, the satellite-related error term, mainly caused by the broadcast ephemeris errors, dominates the stochastic model. By replacing the broadcast ephemeris with IGS precise orbit and clock error products, the ARAIM availability can be significantly improved on a global basis. For ground-based positioning, the receiver-related error term is the intractable one that should be adapted to different receivers and operating environments. Usually, an elevation-dependent function is used to characterize the error. With LSVCE via real data at a station, more accurate coefficients can be obtained, thus improving ARAIM availability at that station.

## 6. Concluding Remarks

In this paper, the authors have proposed refinements to the stochastic model for GNSS ground-based positioning and IM under the ARAIM framework. Refinements to satellite-related, atmosphere-related and receiver-related error terms are presented. Firstly, Gaussian bounds of IGS-precise orbit and clock errors for all GPS satellites are established and used to characterize the satellite-related error term. Secondly, a variable standard deviation is implemented to characterize the residual tropospheric delay after model correction. Thirdly, an elevation-dependent model with adaptive coefficients is used to characterize the receiver-related error term. The corresponding enhancements to ARAIM availability are discussed.

These refinements are evaluated via global simulations and real data experiments. Different schemes are proposed and tested to investigate the enhancement of ARAIM availability for both aviation and ground-based positioning applications. Results show that ARAIM availability can be significantly improved when the satellite-related error term is refined with the use of IGS precise orbit and clock products. With LSVCE via real data at a station, coefficients of the elevation-dependent model can be adapted to different receivers and operating environments, so as to further improve the ARAIM availability at the station. In contrast, PL improvements from using a variable standard deviation to characterize the residual tropospheric delay are smaller, usually on the decimeter or centimeter level. Although these produce marginal improvements to ARAIM availability under current schemes, they would become significant if carrier phase observations are used.

At the first stage, these refinements are designed and evaluated for the SPP mode with the use of GPS dual-frequency IF combination pseudorange observations. Contributions by these investigations to IM can be further extended when using advanced GNSS positioning modes.

## Figures and Tables

**Figure 1 sensors-22-09797-f001:**
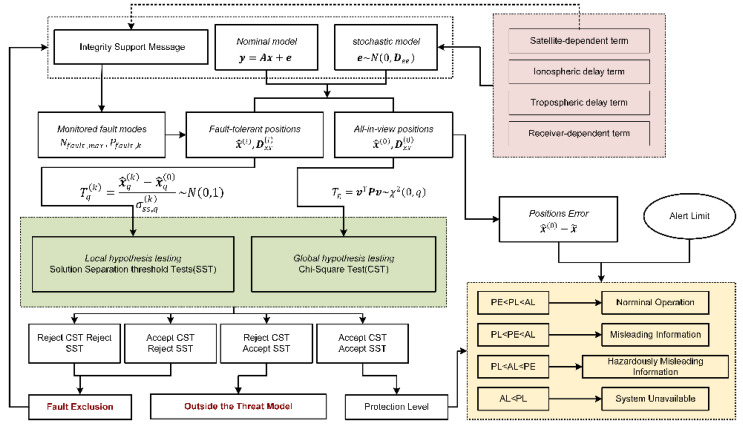
ARAIM algorithm flowchart.

**Figure 2 sensors-22-09797-f002:**
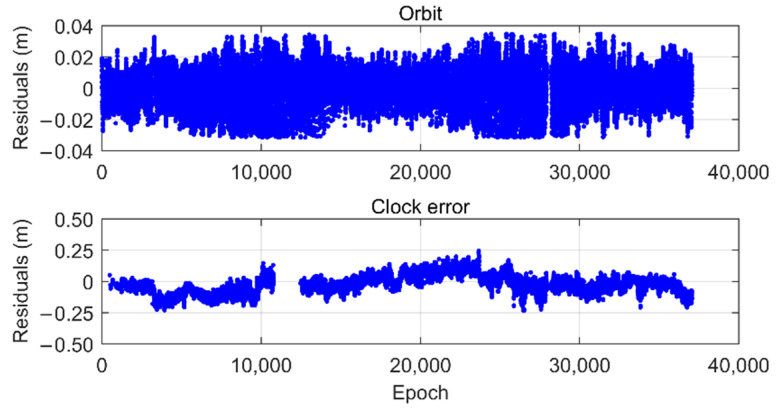
G02 Residuals of orbits and clock errors of the IGC01 products along the LOS direction, referred to the corresponding IGS final products.

**Figure 3 sensors-22-09797-f003:**
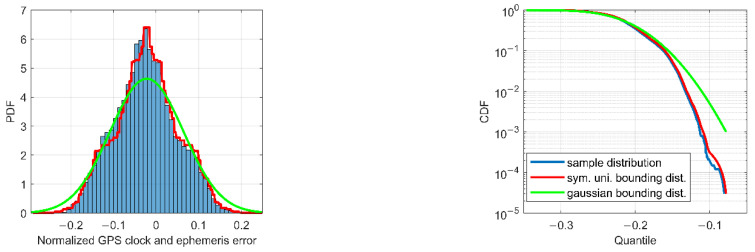
PDF (left panel) and CDF (right) of the samples (blue), the symmetric unimodal bounding distribution (red), and the Gaussian bounding distribution (green).

**Figure 4 sensors-22-09797-f004:**
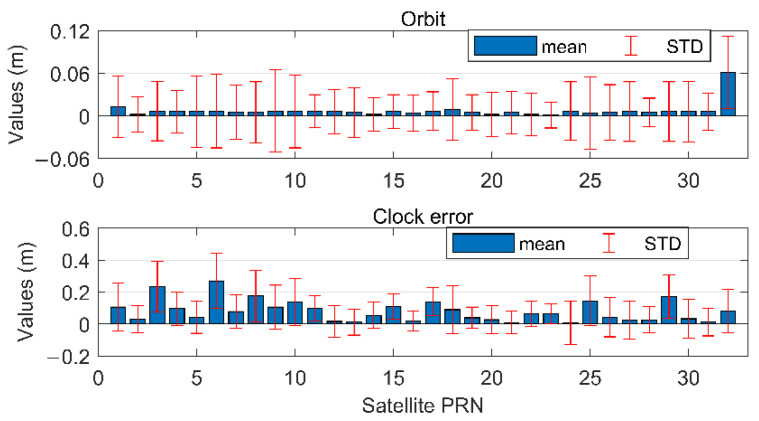
The mean and STD values of the Gaussian bounding distribution for $e_{\mathrm{orbit},\mathrm{RTS}}^{i}$ and ${\tilde{e}}_{\mathrm{clk},\mathrm{RTS}}^{i}$ of all GPS satellite.

**Figure 5 sensors-22-09797-f005:**
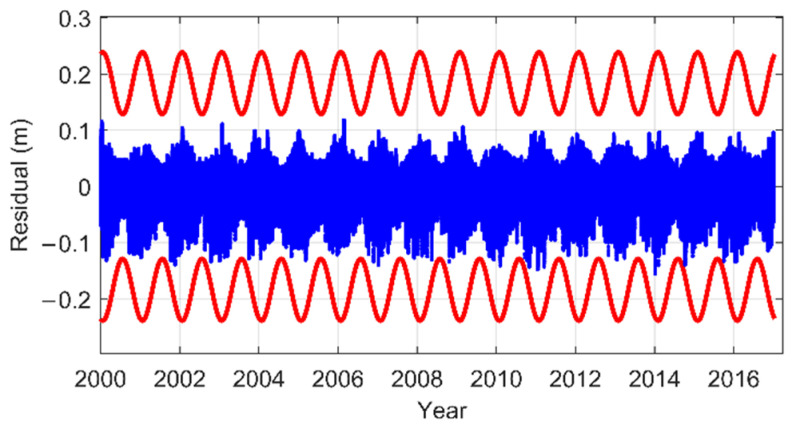
ZHD residuals and the upper bounds of GPT2w in the latitude band 41° N–50° N.

**Figure 6 sensors-22-09797-f006:**
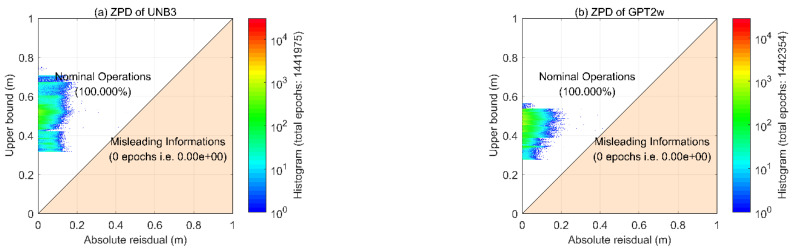
Stanford diagrams for ZPD residuals using UNB3 (**a**) and GPT2w (**b**) models.

**Figure 7 sensors-22-09797-f007:**
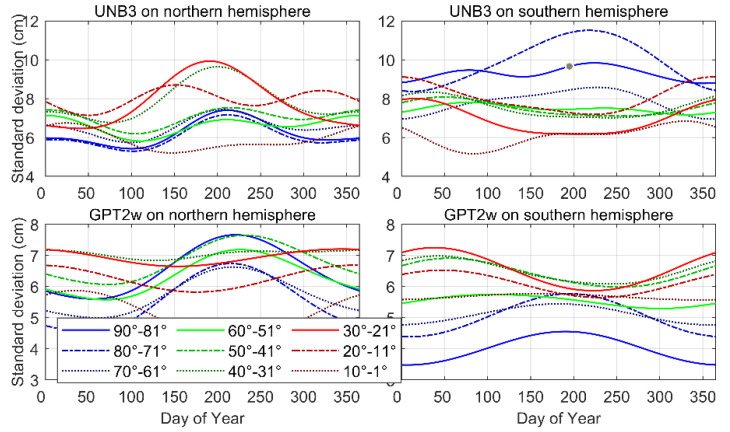
Seasonal variation of $\sigma_{ZPD}$ for UNB3 and GPT2w.

**Figure 8 sensors-22-09797-f008:**
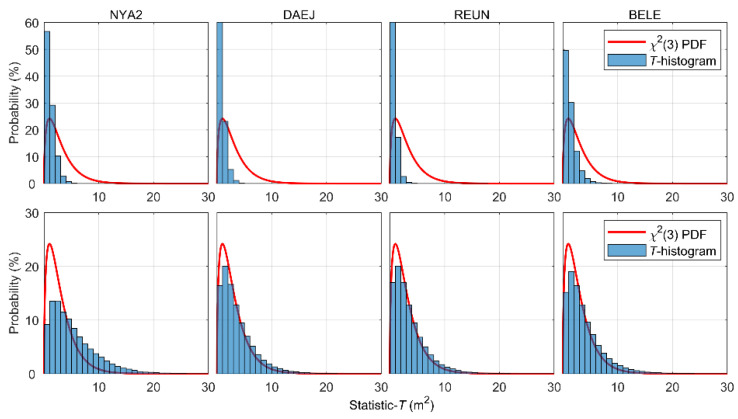
Probability distribution histograms of the test statistic *T* before (upper panels) and after (lower panels) stochastic model refinement for the receiver-related error term.

**Figure 9 sensors-22-09797-f009:**
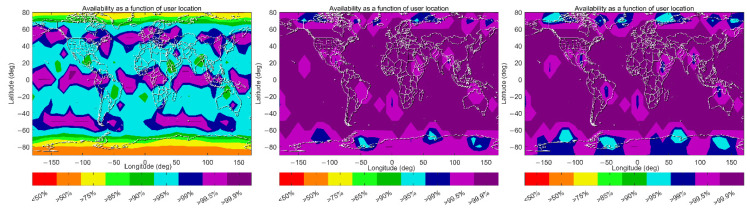
Global coverage of ARAIM availabilities according to the LPV-200 criteria under Schemes A, B and C (left panel: Scheme A, middle panel: Scheme B right panel: Scheme C).

**Figure 10 sensors-22-09797-f010:**
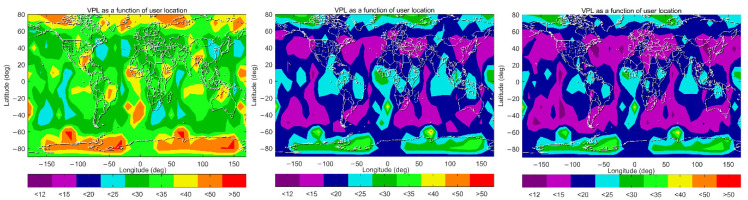
VPL maps (99.5 percentile) under Schemes A–C (left panel: Scheme A, middle panel: Scheme B right panel: Scheme C).

**Figure 11 sensors-22-09797-f011:**
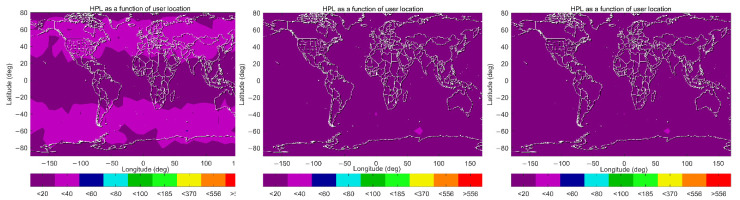
HPL maps (99.5 percentile) under Schemes A–C (left panel: Scheme A, middle panel: Scheme B, right panel: Scheme C).

**Figure 12 sensors-22-09797-f012:**
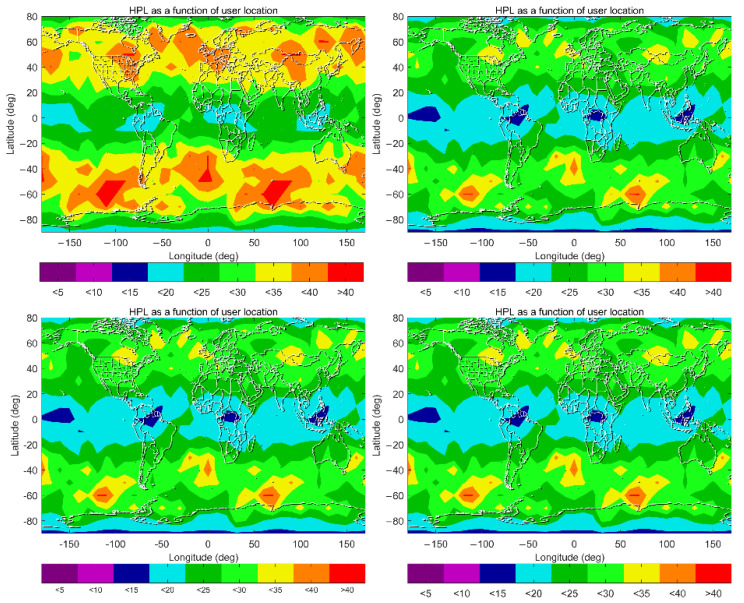
HPL maps (99.5 percentile) under Schemes D–G (top and left panel: Scheme D, top and right panel: Scheme E, bottom and left panel: Scheme F, bottom and right panel: Scheme G).

**Figure 13 sensors-22-09797-f013:**
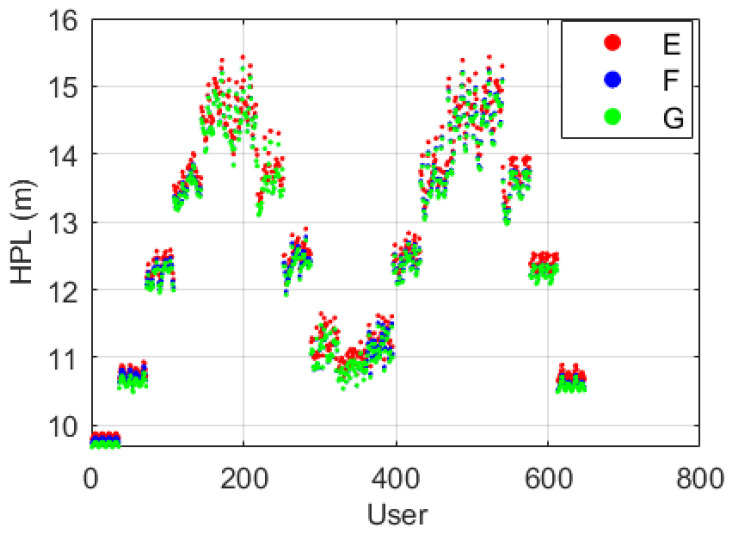
HPL (mean value) under Schemes E–G at each grid point.

**Figure 14 sensors-22-09797-f014:**
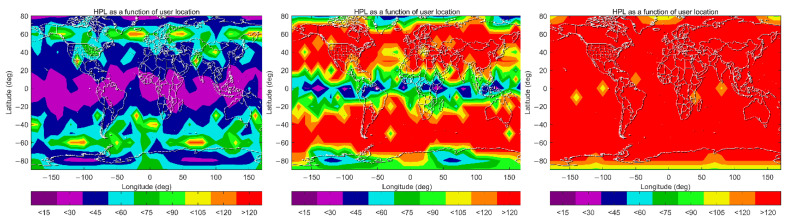
HPL maps (99.5 percentile) under different cut-off elevation angle, (left panel: Scheme H, middle panel: Scheme I right panel: Scheme J).

**Figure 15 sensors-22-09797-f015:**
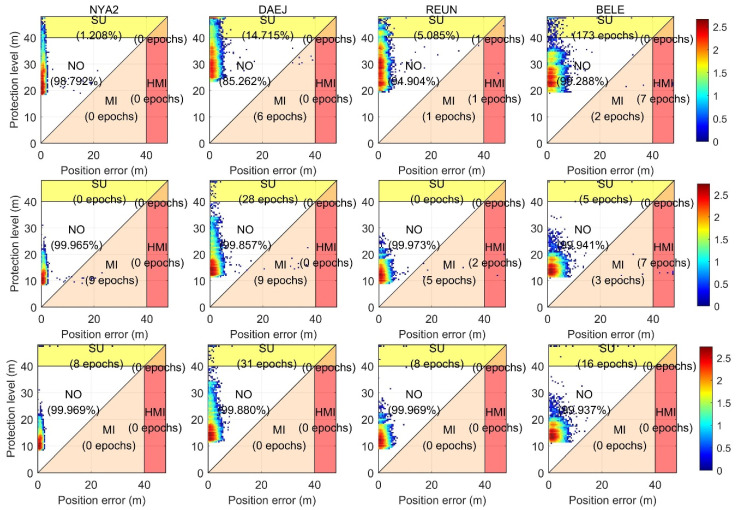
Stanford diagrams of HPEs and HPLs at the four IGS stations before (top) and after (bottom) stochastic model refinement for the receiver-related error term and after fault detection, exclusion (bottom).

**Table 1 sensors-22-09797-t001:** Issues that integrity monitoring should consider in both aviation and ground-based positioning applications.

	Aviation	Ground-Based Positioning
**System**	GPS or GPS/Galileo	BDS, GPS, Galileo, GLONASS
**Frequency**	Single- or dual-	Multiple
**Observation**	Pseudorange	Pseudorange, carrier phase
**Positioning technology**	SPP (IF or CPSP)	SPP, RTK, PPP, PPP-RTK
**Estimation algorithm**	LS	KF
**Satellite effects**	Broadcast satellite ephemerides and clock errors	IGS precise orbits and clock products DCB, UPD products
**Atmosphere effects**	Standard models, i.e., MOPS/EGNOS for troposphere Klobuchar/SBAS correction for ionosphere	Advanced models, i.e., VMF/local accurate model for troposphere IGS GIM/local accurate model for ionosphere
**Receiver effects**	Standard constant models	Vary with hardware, applications, environments
**Fault events**	System based, i.e., satellite or constellation, single or dual faults	Observation based, i.e., outliers in pseudoranges, cycle slips in carrier phase

**Table 2 sensors-22-09797-t002:** Coefficients of elevation-dependent model at four stations.

Station	Latitude (Degree)	Coefficients	
		*a*	*b*
NYA2	78.930	0.484	0.020
DAEJ	36.399	0.635	0.136
REUN	−21.208	0.580	0.144
BELE	−1.409	0.786	0.141

**Table 3 sensors-22-09797-t003:** Detailed designs for parameters under Scheme A to G (unit: m).

	Aviation			Ground-Based Positioning			
	A	B	C	D	E	F	G
$\sigma_{URA,i}$	1	Equation (6)—$\sigma_{URA,i}$		1	Equation (6)—$\sigma_{URA,i}$		
$b_{nom,i}$	0.75	Equation (6)—$m_{URE,i}$		0.75	Equation (6)—$m_{URE,i}$		
$\sigma_{ZPD}$	0.12		Equation (10)—UNB3	0.12		Equation (10)—UNB3	Equation (10)—GPT2w
$\sigma_{user,i}$	Equations (4) and (5)			Equation (20)			

**Table 4 sensors-22-09797-t004:** Numbers of epochs, and numbers (percentages) of unavailable epochs before and after the stochastic model refinement for the receiver-related error term at the four IGS stations.

Station	Number of Epochs	Number of Unavailable Epochs		Number of MI and HMI Epochs	
		Before Refinement	After Refinement	Without FDE	With FDE
NYA2	25,920	313 (1.208%)	0 (0%)	9	0
DAEJ	25,920	3814 (14.715%)	28 (0%)	9	0
REUN	25,920	1318 (5.085%)	0 (0%)	7	0
BELE	25,576	173 (0.667%)	5 (0%)	10	0

## Data Availability

RINEX observation data from IGS stations and the real-time precise satellite orbit and clock products were obtained from the IGS https://igs.org/ (accessed on 1 October 2021). The VMF and ZPD data were obtained from the Vienna Mapping Functions Open Access Data https://vmf.geo.tuwie-n.ac.at/ (accessed on 3 March 2021) and IGS stations, respectively. The open-source software GAMP for GNSS measurement processing is available at https://www.ngs.noaa.gov/gps-toolbox/GAMP.htm (accessed on 10 December 2020) and the open-source software MAAST for ARAIM evaluation is available at https://gps.stanford.edu/resources/software-tosSols/maast (accessed on 20 December 2020).

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
