# Peer review of "ARAIM Stochastic Model Refinements for GNSS Positioning Applications in Support of Critical Vehicle Applications"

_sensors, 2022, doi:10.3390/s22249797_

Round 1

Reviewer 1 Report

The author discusses the challenges for ground-based GNSS positioning technique, and proposes some refinements to solve them. The work is significant. The paper is well orgnized and the expression is satisfactory. Experiments show the excellent results. However, there are some clerical errors needing to be revised carefully.

Reviewer 2 Report

 This paper is on stochastic model suggestion for computing protection level of ground transportation applications. This manuscript include a comprehensive review of ARAIM for aviation and empirical stochastic models, but just switching the model does not guarantee the equal integrity performance to that of aviation. Furthermore, a solution separation, which is an important procedure of the ARAIM, is not considered in this paper, in contrary to the title.

While the stochastic model used in ARAIM is an empirical model based on statistics for a very long period, it is not reasonable to use the suggested model as it can bound a sample presented by the author. As the author mentioned, ground transportation is more challenge application than the aviation because of larger pseudorange outliers and more frequent cycle slips. Nevertheless, the authors chose smaller error models for ground transportation than aviation, and the data selected for the evaluation is obtained from a geodetic receivers, not a low-cost receiver on the road. It is too much obvious that smaller error model results in computing smaller protection levels and accordingly more availability.

Due to the lack of novelty and rationale of the author's claims, it cannot be published at is.

Reviewer 3 Report

This paper gave a sound overview of the ARAIM methods currently applied in aviation, and bring the ARAIM method to the ground vehicles. The paper is organized in the easy readable way. Here are some suggestions.

1. for the ARAIM stochastic model, the refinements should be summarized according to table 1. From the section 4 part, it is not easy to be recognized.

2. In all the Stanford plots, there are no MI or HMIs, according to the regular method for proving the integrity monitoring method, simulated residual observations should be created to show the faulty satellite detection and exclusions. 

3. Since the tile of the paper is as for critical applications, the tolerable HMI should be stated and investigated instead of only error observations in section 4.
